

# Association of healthy lifestyle score with control of hypertension among treated and untreated hypertensive patients: a large cross-sectional study

Ting Dong[1,*], Qin Zhou[2,*], Weiquan Lin[2], Chang Wang[2], Minying Sun[2], Yaohui Li[2], Xiangyi Liu[2], Guozhen Lin[2], Hui Liu[2] and Caixia Zhang[1]

[1] Department of Epidemiology, School of Public Health, Sun Yat-Sen University, Guangzhou, China
[2] Department of Basic Public Health, Guangzhou Center for Disease Control and Prevention, Guangzhou, China
* These authors contributed equally to this work.

Corresponding authors
Hui Liu, gzcdc_liuh@gz.gov.cn
Caixia Zhang,
zhangcx3@mail.sysu.edu.cn

## ABSTRACT

**Background:** Hypertension stands as the leading single contributor to the worldwide burden of mortality and disability. Limited evidence exists regarding the association between the combined healthy lifestyle score (HLS) and hypertension control in both treated and untreated hypertensive individuals. Therefore, we aimed to investigate the association between HLS and hypertension control among adults with treated and untreated hypertension.
**Methods:** This cross-sectional study, including 311,994 hypertension patients, was conducted in Guangzhou using data from the National Basic Public Health Services Projects in China. The HLS was defined based on five low-risk lifestyle factors: healthy dietary habits, active physical activity, normal body mass index, never smoking, and no alcohol consumption. Controlled blood pressure was defined as systolic blood pressure <140 mmHg and diastolic blood pressure <90 mmHg. A multivariable logistic regression model was used to assess the association between HLS and hypertension control after adjusting for various confounders.
**Results:** The HLS demonstrated an inverse association with hypertension control among hypertensive patients. In comparison to the low HLS group (scored 0–2), the adjusted odds ratios (95% confidence intervals) for hypertension were 0.76 (0.74, 0.78), 0.59 (0.57, 0.60), and 0.48 (0.46, 0.49) for the HLS groups scoring 3, 4, and 5, respectively ($P_{trend}$ < 0.001). Notably, an interaction was observed between HLS and antihypertensive medication in relation to hypertension control ($P_{interaction}$ < 0.001). When comparing the highest HLS (scored 5) with the lowest HLS (scored 0–2), adjusted odds ratios (95% confidence intervals) were 0.50 (0.48, 0.52, $P_{trend}$ < 0.001) among individuals who self-reported using antihypertensive medication and 0.41 (0.38, 0.44, $P_{trend}$ < 0.001) among those not using such medication. Hypertensive patients adhering to a healthy lifestyle without medication exhibited better blood pressure management than those using medication while following a healthy lifestyle.
**Conclusion:** HLS was associated with a reduced risk of uncontrolled blood pressure.

## INTRODUCTION

Hypertension stands as the most significant contributor to the global burden of mortality and disability (*Poulter, Prabhakaran & Caulfield, 2015*). Between 1990 and 2019, the number of individuals aged 30–79 with high blood pressure doubled, reaching around 626 million females and 652 million males living with hypertension worldwide by 2019 (*(NCD-RisC) NRFC, 2021*). Approximately 75% of people with hypertension reside in low- and middle-income nations (*Mills et al., 2016*). The prevalence of hypertension among Chinese adults has consistently increased with urbanization and the aging of the population (*Wang et al., 2018*). Despite hypertension management being a national public health priority, the blood pressure control rate remains low, estimated at 7.2% in China from 2014 to 2017 (*Lu et al., 2017*). Identifying effective strategies for hypertension management is crucial.

Various aspects of individual lifestyle factor, such as dietary habits, physical activity, body mass index (BMI), smoking, and alcohol consumption, have been independently demonstrated to influence blood pressure levels. Studies suggests that maintaining a healthy dietary habits, engaging in regular exercise, maintaining a normal BMI, never smoking, and reducing alcohol consumption are beneficial for hypertension control (*Chudek et al., 2021*; *Lopes et al., 2020*; *Niu et al., 2021*; *Ozemek et al., 2018*; *Roerecke et al., 2017*). However, these studies have ignored the fact that multiple lifestyle-related factors often coexist in individuals, collectively affecting people's physical health (*Aleksandrova et al., 2014*; *Erben et al., 2019*). Thus, it is essential to consider these lifestyle factors simultaneously, along with their combined consequences.

Previous studies conducted across diverse nations consistently indicate that adherence to healthy lifestyle practices is associated with a decreased risk of various diseases, including hypertension, type two diabetes, early-onset dementia, bladder cancer, and cardiometabolic multimorbidity (*He et al., 2023a*, *2023b*; *Li et al., 2023*; *Xie et al., 2022*; *Zhen et al., 2023*). Notably, studies conducted in China, Iran, Japan, and America have demonstrated the association between a healthy lifestyle and reduced blood pressure (*Akbarpour et al., 2018*; *Appel et al., 2003*; *Xiao et al., 2020*; *Yokokawa et al., 2014*). However, the sample sizes of studies exploring the relationship between a healthy lifestyle and hypertension control vary from a few hundred to a few thousand, and there is a lack of studies in large sample populations.

The effective management of hypertension often relies on the pivotal role of antihypertensive medication treatment. However, the use of anti-hypertensive drugs may lead to decreased adherence to healthy lifestyles (*Neutel & Campbell, 2008*). So far, some studies have evaluated the relationship between combined lifestyle factors and hypertension control in individuals undergoing treatment (*Cherfan et al., 2020*; *Yokokawa et al., 2014*). The association between lifestyle factors and blood pressure control has also been explored in untreated hypertensive patients (*Appel et al., 2003*). However, few studies have simultaneously evaluated the association between lifestyle behavior and hypertension control among both treated and untreated hypertensive patients (*Akbarpour et al., 2018*),

which could be an important step in blood pressure management. To our knowledge, only a cross-sectional study in Iran has investigated the relationship between a healthy lifestyle and hypertension control among those aware and using antihypertensive medications and those aware but not taking medication (*Akbarpour et al., 2018*). However, this study had a limited sample size and did not include individuals aged over 65 years. Additionally, there are no related studies in China.

To address these knowledge gaps, our objective was to examine the association of a healthy lifestyle score (HLS), encompassing dietary habits, physical activity, BMI, smoking, and alcohol consumption, with hypertension control among both treated and untreated hypertensive patients in China.

## METHODS

### Study population

The study enrolled hypertensive patients receiving the National Basic Public Health Services in Guangzhou in 2018. The National Basic Public Health Services projects represent the fundamental healthcare services provided by the government free of charge to all residents, addressing primary health issues among urban and rural populations. These services primarily focus on children, pregnant women, the elderly, and patients with chronic diseases (*Wang et al., 2019*).

A total of 375,912 hypertensive patients were enrolled in this study. The inclusion criteria were as follows: aged ≥35 years, Guangdong natives or residents who had lived in Guangzhou for at least six months, primary hypertension diagnosis, and undergoing a physical examination. Hypertension was defined as systolic blood pressure (SBP) ≥140 mmHg or diastolic blood pressure (DBP) ≥90 mmHg, or reporting current use of antihypertensive medication, or self-reported hypertension (*Cao et al., 2021*). Exclusion criteria included repeat participants ($n = 13,284$); individuals with missing or incomplete blood pressure information ($n = 10,343$); those with missing or incomplete lifestyle factor information ($n = 11,079$), participants with extreme blood pressure values ($n = 20,672$), and individuals with extreme values in lifestyle factors ($n = 8,540$). Ultimately, 311,994 participants were included in the final data analysis (Fig. 1). Table S1 compares the characteristics of both the excluded participants and those included in the study. Extreme values were defined as below the first quartile (Q1) minus 1.5 times the interquartile range or above the third quartile (Q3) plus 1.5 times interquartile range (*Haileslasie et al., 2020*).

This research was conducted in accordance with the ethical standards outlined in the 1964 Declaration of Helsinki and received approval from the Ethics Committee of the School of Public Health at Sun Yat-Sen University (approval number: 2023-007).

### Data collection

All participants underwent face-to-face interviews conducted by trained medical staff at community-level health facilities using a structured questionnaire. The collected information included socio-demographic characteristics (age, sex, ethnicity, educational level, marital status), lifestyle factors (dietary habits, physical activity, smoking, and alcohol consumption), history of diabetes, family history of hypertension, and medication details.

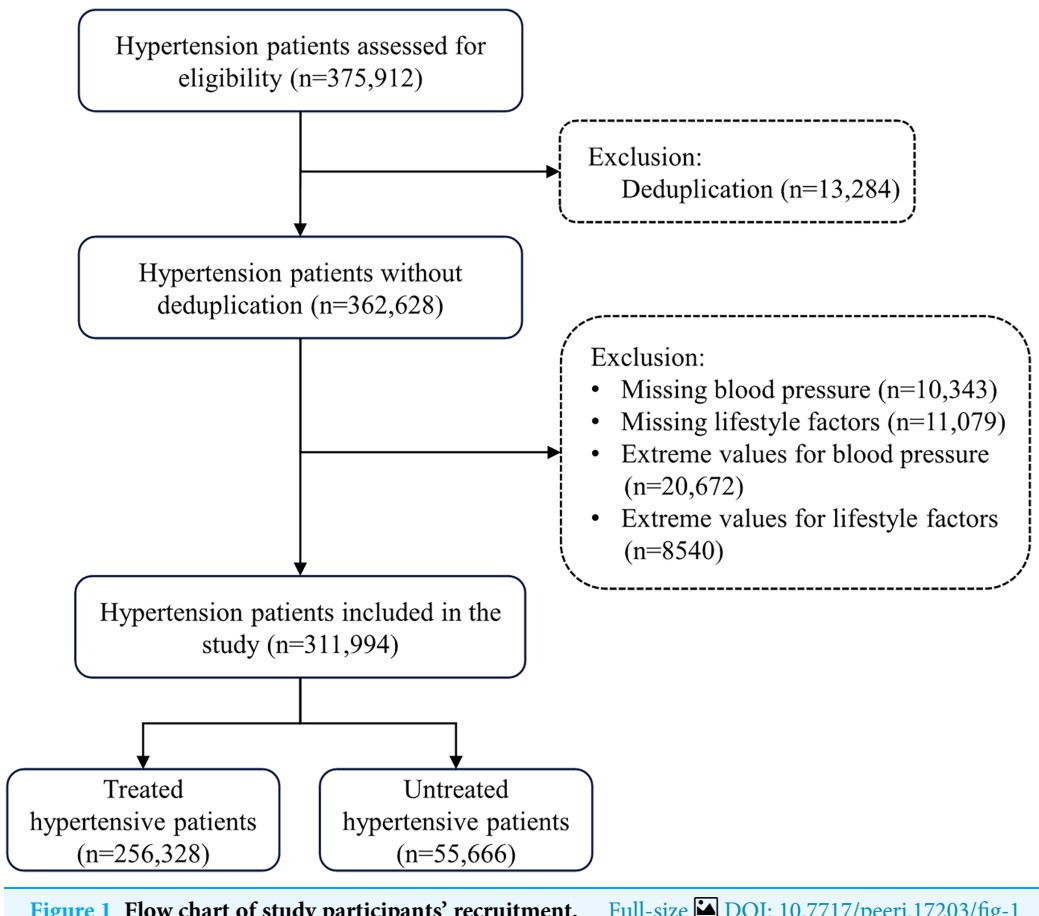

**Figure 1 Flow chart of study participants' recruitment.**

Dietary habits were divided into balanced meat and vegetable, meat-based, vegetarian-based, salt-loving, oil-loving, and sugar-loving. Physical activity levels were assessed based on exercise frequency, categorized as never, occasional, more than once a week, and daily. Occasional and more than once a week were calculated based on once a week and three times a week, respectively. Smoking status was classified as never smokers and smokers (former and current smokers). Never smokers were participants who reported never smoking, while smokers included participants who reported current smoking and ever smoking (*Yao et al., 2019*). Similarly, drinking status was divided into never drinkers and drinkers (ever and current drinkers). Never drinkers were defined as participants who reported never drinking, while drinkers included participants who reported current drinking and former drinking (*Zhao et al., 2021*).

Standardized methods were used to measure participants' height and weight. BMI was calculated by dividing body weight (in kilograms) by the square of height (in meters). According to the Working Group on Obesity in China, underweight, normal weight, overweight, and obesity were defined as BMI <18.5, 18.5–23.9, 24.0–27.9, and ≥28.0 kg/m$^2$, respectively (*Yuan et al., 2017*).

Blood pressure was assessed with a single measurement, averaging readings from the left and right arms. The measurement utilized upper-arm electronic medical blood pressure

**Table 1 Components of lifestyle score.**

| Lifestyle factors | Category | Healthy/ Unhealthy | Score |
|---|---|---|---|
| Dietary habits | Balanced consumption of meat and vegetable, moderate salt consumption, moderate oil consumption, and moderate added sugar consumption | Healthy | 1 |
| | Prefer meat/prefer vegetarian diet/high salt consumption/high oil consumption/high added sugar consumption | Unhealthy | 0 |
| Physical activities | Exercise frequency ≥2 times weekly | Healthy | 1 |
| | Exercise frequency <2 times weekly | Unhealthy | 0 |
| Body mass index | $18.5–23.9 \text{ kg/m}^2$ | Healthy | 1 |
| | $<18.5 \text{ kg/m}^2/≥24.0 \text{ kg/m}^2$ | Unhealthy | 0 |
| Smoking | Never | Healthy | 1 |
| | Current/former | Unhealthy | 0 |
| Alcohol consumption | Never | Healthy | 1 |
| | Current/former | Unhealthy | 0 |

monitors certified according to International Standard Protocols (*Xu et al., 2022*). Systolic and diastolic blood pressure were measured in the seated position for both arms. The mean value of the two measurements from the left and right arms was considered as an individual's blood pressure measurements. Controlled hypertension was defined as mean SBP <140 mmHg and mean DBP <90 mmHg (*Kimani et al., 2019*). Uncontrolled hypertension was defined as mean SBP ≥140 mmHg and/or mean DBP ≥90 mmHg (*Yokokawa et al., 2011*).

## Definition of healthy lifestyle score

Drawing upon previous knowledge and contemporary public health recommendations, we formulated a HLS using information regarding five lifestyle factors: dietary habits, physical activity, BMI, smoking, and alcohol consumption (*Fukunaga et al., 2020*; *Ye et al., 2021*). Participants were assigned one point for each of the following protective lifestyle factors: healthy dietary habits (balanced meat and vegetable, moderate salt consumption, moderate oil consumption, and moderate added sugar consumption) (*Che et al., 2023*), active physical activity (exercise frequency ≥2 times weekly) (*Yokokawa et al., 2014*), normal BMI ($18.5–23.9 \text{ kg/m}^2$) (*Wang et al., 2022*), never smoking (*Li et al., 2021*), and no alcohol consumption (*Che et al., 2023*). The details of the lifestyle score are outlined in Table 1. These five scores were summed to generate a final low-risk lifestyle score ranging from 0 to 5, with lower scores indicating a less healthy lifestyle.

## Statistical analysis

We divided the HLS into five groups (0–2, 3, 4, and 5). Due to the small number of participants with HLS scores of 0 and 1 ($n = 263$ and 4,886, respectively), we combined the three lowest categories, considering those with HLS of 0–2 as the reference category in all analyses. Categorical variables are expressed as numbers (percentages). Basic characteristics of controlled and uncontrolled hypertension were compared using

chi-square tests for categorical variables. A multivariate logistic regression model was used to estimate odds ratio (OR) and 95% confidence intervals (CI) for the analysis of the relationship between HLS and hypertension control, with participants following an unhealthy lifestyle as the reference. The multivariable model was adjusted for all potential confounders, including age, sex, ethnicity, educational level (primary school or below, junior high school, senior high school, college or above, and unknown), marital status, history of diabetes, family history of hypertension, and self-reported using antihypertensive medication.

Stratified analyses were conducted by antihypertensive drugs use (reporting no use of antihypertensive medication *vs.* reported using antihypertensive medication). The significance of differences in effects between subgroups was tested using the formula: $z = (\beta_1 - \beta_2)\sqrt{(SE_1)^2 + (SE_2)^2}$, where $\beta_1$ and $\beta_2$ indicate the estimates for each subgroup, and $SE_1$ and $SE_2$ represent their respective standard errors (*Altman & Bland, 2003*). Interaction refers to the variations in the response magnitude observed across different levels of a specific factor (*McCabe et al., 2022*). We calculated it by including interaction terms in the multiple regression model. To assess the association of each component of the HLS with hypertension control, we constructed univariate adjustment models controlling for the above covariates, as well as the other components of the HLS for each component.

All statistical analysis were performed using R x64 4.1.1. Statistical significance was determined at a two-tailed *P* value of less than 0.05.

## RESULTS

### Baseline characteristics

As shown in Table 2, of the 311,994 participants, 243,101 (77.92%) were 65 years and older, 124,116 (39.78%) were males, and 256,328 (82.16%) reported using antihypertensive medication. In comparison to treated patients, untreated individuals were more likely to be older, exhibit higher proportions of males, and have lower educational attainment. Participants reporting no use of antihypertensive medication were also more likely to have a lower BMI, engage in less physical activity, alcohol consumption, possess a lower HLS, report a lower history of diabetes, and have less family history of hypertension than those who reported using antihypertensive medication. There was no significant difference in ethnicity ($P = 0.572$), marital status ($P = 0.195$), dietary habits ($P = 0.214$), and smoking ($P = 0.117$).

### Association of HLS with hypertension control

Table 3 shows the association of HLS with hypertension control. In comparison to patients with an HLS score of 0–2, the multivariable adjusted ORs and 95% CIs for those with a score of 3, 4, and 5 were 0.76 (0.74, 0.78), 0.59 (0.57, 0.60), and 0.48 (0.46, 0.49; $P_{trend} < 0.001$), respectively. Stratified analysis revealed an interaction between self-reported using antihypertensive medication and HLS concerning hypertension control ($P_{interaction} < 0.001$). Comparing the highest HLS (scored 5) with the lowest HLS (scored

**Table 2 Baseline characteristics according to the treatment status of hypertension.**

| | Total, No. (%) | Treated, No. (%) | Untreated, No. (%) | P value |
|---|---|---|---|---|
| Overall | 311,994 (100) | 256,328 (82.16) | 55,666 (17.84) | |
| Age, years | | | | <0.001 |
| <65 | 68,893 (22.08) | 57,122 (22.28) | 11,771 (21.15) | |
| ≥65 | 243,101 (77.92) | 199,206 (77.72) | 43,895 (78.85) | |
| Sex | | | | <0.001 |
| Males | 124,116 (39.78) | 101,569 (39.62) | 22,547 (40.50) | |
| Females | 187,878 (60.22) | 154,759 (60.38) | 33,119 (59.50) | |
| Ethnicity | | | | 0.572 |
| Han | 311,497 (99.84) | 255,925 (99.84) | 55,572 (99.83) | |
| Others | 497 (0.16) | 403 (0.16) | 94 (0.17) | |
| Educational level | | | | <0.001 |
| Primary school or below | 92,067 (29.51) | 72,494 (28.28) | 19,573 (35.16) | |
| Junior high school | 63,277 (20.28) | 53,573 (20.90) | 9,704 (17.43) | |
| Senior high school/Secondary technical school | 69,056 (22.13) | 59,082 (23.05) | 9,974 (17.92) | |
| College or above | 85,857 (27.52) | 69,782 (27.22) | 16,075 (28.88) | |
| Unknown | 1,737 (0.56) | 1,397 (0.55) | 340 (0.61) | |
| Marital status | | | | 0.195 |
| Married | 275,026 (88.15) | 226,046 (88.19) | 48,980 (87.99) | |
| Others | 36,968 (11.85) | 30,282 (11.81) | 6,686 (12.01) | |
| BMI, kg/m$^2$ | | | | <0.001 |
| <18.5 | 7,028 (2.25) | 5,291 (2.06) | 1,737 (3.12) | |
| 18.5–23.9 | 138,602 (44.42) | 111,663 (43.56) | 26,939 (48.39) | |
| 24.0–27.9 | 123,647 (39.63) | 103,025 (40.19) | 20,622 (37.05) | |
| ≥28.0 | 42,717 (13.69) | 36,349 (14.18) | 6,368 (11.44) | |
| Active physical activity | 183,534 (58.83) | 154,495 (60.27) | 29,039 (52.17) | <0.001 |
| Healthy dietary habits | 303,965 (97.43) | 249,689 (97.41) | 54,276 (97.50) | 0.214 |
| Smoking | 44,523 (14.27) | 36,697 (14.32) | 7,826 (14.06) | 0.117 |
| Alcohol consumption | 27,937 (8.95) | 22,579 (8.81) | 5,358 (9.63) | <0.001 |
| HLS | | | | <0.001 |
| 0–2 | 24,863 (7.97) | 20,230 (7.89) | 4,633 (8.32) | |
| 3 | 81,284 (26.05) | 66,223 (25.84) | 15,061 (27.06) | |
| 4 | 139,772 (44.80) | 114,940 (44.84) | 24,832 (44.61) | |
| 5 | 66,075 (21.18) | 54,935 (21.43) | 11,140 (20.01) | |
| History of diabetes | 71,246 (22.84) | 64,486 (25.16) | 6,760 (12.14) | <0.001 |
| Family of hypertension history | 37,355 (11.97) | 34,145 (13.32) | 3,210 (5.77) | <0.001 |

Note:
   BMI, body mass index; HLS, healthy lifestyle score.

0–2), the adjusted OR (95% CIs) was 0.50 (0.48, 0.52, $P_{\text{trend}} < 0.001$) in individuals reporting no use of antihypertensive medication and 0.41 (0.38, 0.44, $P_{\text{trend}} < 0.001$) in those who reported using antihypertensive medication (Table 4). Compared to individuals who self-reported using antihypertensive medication, those not using such medication exhibited stronger associations between HLS and the risk of uncontrolled hypertension (all

**Table 3 Association of healthy lifestyle score with hypertension control.**

| | Healthy lifestyle score | | | | $P_{trend}$ |
|---|---|---|---|---|---|
| | 0–2 | 3 | 4 | 5 | |
| **Total** | | | | | |
| No of patients | 24,863 | 81,284 | 139,772 | 66,075 | |
| Model 1[a] | 1.00 | 0.84 [0.82, 0.87] | 0.66 [0.64, 0.67] | 0.53 [0.52, 0.55] | <0.001 |
| Model 2[b] | 1.00 | 0.76 [0.74, 0.78] | 0.59 [0.57, 0.60] | 0.48 [0.46, 0.49] | <0.001 |
| **Untreated** | | | | | |
| No. of patients | 4,633 | 15,061 | 24,832 | 11,140 | |
| Model 1[a] | 1.00 | 0.72 [0.68, 0.77] | 0.55 [0.52, 0.59] | 0.46 [0.43, 0.49] | <0.001 |
| Model 2[c] | 1.00 | 0.65 [0.61, 0.70] | 0.49 [0.46, 0.52] | 0.41 [0.38, 0.44] | <0.001 |
| **Treated** | | | | | |
| No. of patients | 20,230 | 66,223 | 114,940 | 54,935 | |
| Model 1[a] | 1.00 | 0.87 [0.85, 0.90] | 0.68 [0.66, 0.71] | 0.55 [0.53, 0.57] | <0.001 |
| Model 2[c] | 1.00 | 0.79 [0.76, 0.81] | 0.61 [0.59, 0.63] | 0.50 [0.48, 0.52] | <0.001 |

Notes:
[a] Unadjusted OR (95% CI).
[b] OR was adjusted for age, sex, ethnicity, educational level, marital status, history of diabetes, family history of hypertension, and self-reported using antihypertensive drugs.
[c] OR was adjusted for age, sex, ethnicity, educational level, marital status, history of diabetes, and family history of hypertension.

**Table 4 Association between healthy lifestyle score and hypertension control stratified by treatment status of hypertension.**

| Healthy lifestyle score | | Treatment status of hypertension | |
|---|---|---|---|
| | | Treated | Untreated |
| 0–2 | OR (95% CI) | 1.00 | 1.00 |
| 3 | OR (95% CI) | 0.79 [0.76, 0.81] | 0.65 [0.61, 0.70] |
| | $P$[a] | <0.001 | |
| 4 | OR (95% CI) | 0.61 [0.59, 0.63] | 0.49 [0.46, 0.52] |
| | $P$[a] | <0.001 | |
| 5 | OR (95% CI) | 0.50 [0.48, 0.52] | 0.41 [0.38, 0.44] |
| | $P$[a] | <0.001 | |
| | $P_{interaction}$ | <0.001 | |

Notes:
OR was adjusted for age, sex, ethnicity, educational level, marital status, history of diabetes, and family history of hypertension.
[a] $P$ value for difference between the treated group and the untreated group, is given by two-sample z-test: $z = (\beta_1 - \beta_2)\sqrt{(SE_1)^2 + (SE_2)^2}$, where $\beta_1$ and $\beta_2$ indicate the estimates for each subgroup, and $SE_1$ and $SE_2$ represent their respective standard errors.

$P$ for comparison <0.001). Hypertensive patients adhering to a healthy lifestyle without medication demonstrated better blood pressure management than those who used medication while following a healthy lifestyle.

## Association of individual lifestyle factors with hypertension control

The associations between individual lifestyle factors, HLS, and controlled hypertension are described in Fig. 2. Healthy dietary habits, active physical activity, normal BMI, never

| Lifestyle factors | No. of Events/Total | | OR (95%CI) |
|---|---|---|---|
| **All patients** | | | |
| Active physical activity | 183,534/311,994 | | 0.89 (0.88, 0.91) |
| Never smoking | 267,471/311,994 | | 0.80 (0.78, 0.82) |
| Healthy dietary habits | 303,965/311,994 | | 0.78 (0.74, 0.82) |
| No alcohol consumption | 284,057/311,994 | | 0.75 (0.73, 0.77) |
| Normal body mass index | 138,602/311,994 | | 0.73 (0.72, 0.74) |
| HLS | 66,075/311,994 | | 0.48 (0.46, 0.49) |
| **Treated patients** | | | |
| Active physical activity | 154,495/256,328 | | 0.88 (0.87, 0.90) |
| Never smoking | 219,631/256,328 | | 0.83 (0.80, 0.85) |
| Healthy dietary habits | 249,689/256,328 | | 0.80 (0.76, 0.84) |
| No alcohol consumption | 233,749/256,328 | | 0.75 (0.73, 0.78) |
| Normal body mass index | 111,663/256,328 | | 0.74 (0.72, 0.75) |
| HLS | 54,935/256,328 | | 0.50 (0.48, 0.51) |
| **Untreated patients** | | | |
| Active physical activity | 29,039/55,666 | | 0.91 (0.88, 0.94) |
| No alcohol consumption | 50,308/55,666 | | 0.73 (0.68, 0.78) |
| Healthy dietary habits | 54,276/55,666 | | 0.71 (0.64, 0.79) |
| Never smoking | 47,840/55,666 | | 0.71 (0.67, 0.75) |
| Normal body mass index | 26,939/55,666 | | 0.70 (0.68, 0.73) |
| HLS | 11,140/55,666 | | 0.41 (0.38, 0.44) |

0.3 0.4 0.5 0.6 0.7 0.8 0.9 1.0

**Figure 2 Association between controlled hypertension and individual lifestyle factors and HLS.** Dietary habits, physical activity, body mass index, smoking and alcohol consumption factors with a score of 0 and HLS with a score of 0–2 were used as reference groups. Adjusted for age, sex, ethnicity, educational level, marital status, history of diabetes, family history of hypertension, self-reported using antihypertensive drugs, as well as other components of healthy lifestyle score. Stratified factors were not adjusted in each model; HLS, healthy lifestyle score.

smoking, and no alcohol consumption were independently associated with a negative correlation with blood pressure control. Moreover, individuals who adopted all five healthy lifestyle factors exhibited better blood pressure control compared to those adopting only a single factor. Similarly, subgroup analysis by antihypertensive drugs use showed that healthy dietary habits, active physical activity, normal BMI, never smoking and no alcohol consumption were associated with well-controlled blood pressure.

## DISCUSSION

The results indicated an inverse association between HLS and hypertension control among hypertensive patients. Hypertensive patients adhering to a healthy lifestyle without taking

medication demonstrated better blood pressure management than those who used medication and followed a healthy lifestyle.

We observed a strong inverse association between HLS and hypertension control. Our findings align with the results of *Akbarpour et al. (2018)* who reported that the risk of uncontrolled hypertension in individuals with an unhealthy lifestyle was approximately 37% higher than in those adhering to a moderate lifestyle. A cross-sectional study conducted in France revealed that modifiable unhealthy lifestyle factors were associated with an increased risk of uncontrolled hypertension, particularly in treated hypertensive subjects (*Cherfan et al., 2020*). The FRESH study in Japan reported that maintaining a healthy lifestyle served as a protective factor for blood pressure management (*Yokokawa et al., 2011*, *2014*). In a Chinese Community Intervention Trial, the rate of blood pressure control improved by 56.1% following a one-year lifestyle intervention (*Xiao et al., 2020*). Additionally, in the PREMIER clinical trial involving 810 participants with nonoptimal blood pressure, a 6-month lifestyle intervention comprising weight loss, sodium restriction, enhanced physical activity, alcohol intake restriction, and improved diet quality significantly reduced SBP by 4.3 mmHg (*Appel et al., 2003*). Our findings align with this existing body of evidence, confirming that a higher HLS is linked to a reduced risk of uncontrolled hypertension. These cumulative findings underscore the pivotal role of lifestyle factors in the successful management of hypertension. Therefore, integrating this combined evidence can inform the development and implementation of impactful lifestyle intervention strategies, offering potential advancements in hypertension management.

Our study revealed that 82.16% of hypertensive patients were self-reported using antihypertensive medication. Patients who used medication and adhered to a healthy lifestyle exhibited worse blood pressure control than those who adhered to a healthy lifestyle without medication. Several possible explanations exist for the poorer blood pressure control in individuals adhering to a healthy lifestyle while using antihypertensive drugs. Firstly, in our study, the prevalence of diabetes and family history of hypertension was higher in hypertensive patients using medication than in those not using medication. Secondly, individuals not taking medication may have lower blood pressure levels than those taking medication (*Akbarpour et al., 2018*). Lastly, the lower adherence among people taking medication may contribute to uncontrolled hypertension (*Macquart de Terline et al., 2020*). To our knowledge, only one study has concurrently examined the relationship between HLS and blood pressure control among treated and untreated hypertensive patients (*Akbarpour et al., 2018*). A cross-sectional study in Iran, including 2,577 participants with hypertension, found that the risk of uncontrolled hypertension in individuals with good lifestyle behaviors was 1% and 45% lower than those with poor lifestyle behaviors among treated and untreated hypertensive patients, respectively. However, the study's findings were not statistically significant, possibly due to the relatively small sample size (*Akbarpour et al., 2018*). Additionally, the Dongfeng-Tongji cohort study in China demonstrated that patients adhering to a healthy lifestyle but not using medications had a lower mortality rate than patients using drugs and adhering to a healthy lifestyle (*Lu et al., 2022*). This highlights the significance of promoting a healthy lifestyle, especially for individuals whose blood pressure remains inadequately controlled despite

the use of antihypertensive medications. In managing hypertensive patients, healthcare professionals should prioritize lifestyle interventions, complementing pharmacological management, to improve blood pressure control, particularly for those with suboptimal outcomes even with antihypertensive drug therapy.

The analysis investigating blood pressure control in relation to individual lifestyle factors indicated that healthy dietary habits, active physical activity, normal BMI, never smoking, and no alcohol consumption were independently and inversely correlated with uncontrolled hypertension. These findings align with previous studies examining the associations between individual lifestyle factors and hypertension control (*Börjesson et al., 2016*; *Foti et al., 2022*; *Kudo et al., 2015*; *Qin et al., 2021*; *Yang et al., 2022*). The present study contributes to the evidence that healthy dietary habits, active physical activity, normal BMI, and never smoking, and no alcohol consumption contribute to blood pressure management. Our findings also demonstrated that individuals who adopted all five healthy lifestyle factors had better blood pressure control than those who adopted only a single lifestyle factor. A study involving 1,018 Irish adults revealed that individuals with four or more protective lifestyles had a lower risk of developing hypertension than those with a single lifestyle (*Villegas, Kearney & Perry, 2008*). The Aerobics Center Longitudinal Study found that individuals with only a single lifestyle were more likely to develop hypertension than those with five healthy lifestyles (*Banda et al., 2010*). This study, along with previous studies, indicates that the HLS serves as a composite indicator, capturing an individual's overall commitment to a healthy lifestyle. It emphasizes the cumulative impact of multiple healthy lifestyle factors on blood pressure management.

Our study possesses certain strengths. Firstly, a key advantage of this study lies in its extensive sampling of Chinese adults with hypertension. Furthermore, the investigation explores the relationship between HLS and blood pressure control across both treated and untreated hypertensive patient groups.

Nevertheless, this study is subject to several limitations. Firstly, a large number of participants were excluded due to incomplete data, which could lead to selection bias and impact the generalizability of our findings. The findings may be more applicable to middle-aged and elderly patients with hypertension. Secondly, the potential for recall bias exists due to self-reported lifestyle factors. Despite efforts to minimize this bias through professional training of health workers, it cannot be completely ruled out. Thirdly, the reliability and validity of the HLS were not evaluated. Future studies could conduct reliability tests or focus on direct assessments of diet, physical activity, and other lifestyle factors, alongside BMI, to enhance the accuracy and reliability of the HLS. Fourthly, the definition of healthy dietary habits relied on simple dietary habits due to a lack of data on nutrient intake. The evaluation of various nutrient intakes, particularly sodium, which significantly influences blood pressure, was not possible. Therefore, our definition of healthy dietary habits may be specific to our study population, and generalization of results could be limited. However, a cohort study in Japan also defined a healthy eating factor based on dietary habits (*Yokokawa et al., 2014*). Fifthly, the inherent reverse-causality bias in cross-sectional analyses prevents confirmation of a causal association between a healthy lifestyle and controlled hypertension. Sixthly, although we addressed confounding bias

through adjustments in the multivariate mixture model and stratified analysis. Residual confounders may persist since certain critical variables, such as hypertension duration and medication adherence, were not considered. Seventhly, our definition of controlled blood pressure relies on a single measurement rather than the average of multiple measurements. However, we measured blood pressure in both arms and averaged the readings. Lastly, as our study is cross-sectional, information on lifestyle and medications was collected only once at baseline. The investigation of the relationship between changes in lifestyle, medication utilization, and blood pressure control would require prospective studies.

## CONCLUSIONS

This study demonstrated that a healthier lifestyle was associated with a reduced risk of uncontrolled blood pressure. Additionally, individuals who adhered to a healthy lifestyle without taking medication exhibited better blood pressure management compared to those using medication while following a healthy lifestyle. Adoption of a healthy lifestyle is critical for improving blood pressure levels, especially in those with inadequate blood pressure control despite self-reported using antihypertensive drug. The results of this study could draw public attention to lifestyles factors and offer evidence for the management of hypertension, guiding the development of prevention and therapeutic strategies. Additional prospective studies are warranted to confirm our findings and assess the impact of lifestyle changes on blood pressure control in individuals with hypertension.

## ACKNOWLEDGEMENTS

We gratefully acknowledge the participation and contribution of the participants; without them, this study would not have been possible.

### Funding
The authors received no funding for this work.

### Competing Interests
The authors declare that they have no competing interests.

### Author Contributions
- Ting Dong conceived and designed the experiments, analyzed the data, prepared figures and/or tables, authored or reviewed drafts of the article, and approved the final draft.
- Qin Zhou performed the experiments, analyzed the data, prepared figures and/or tables, authored or reviewed drafts of the article, collected the data, and approved the final draft.
- Weiquan Lin performed the experiments, prepared figures and/or tables, and approved the final draft.
- Chang Wang performed the experiments, prepared figures and/or tables, and approved the final draft.
- Minying Sun performed the experiments, prepared figures and/or tables, and approved the final draft.

- Yaohui Li performed the experiments, prepared figures and/or tables, and approved the final draft.
- Xiangyi Liu performed the experiments, prepared figures and/or tables, and approved the final draft.
- Guozhen Lin performed the experiments, prepared figures and/or tables, and approved the final draft.
- Hui Liu conceived and designed the experiments, performed the experiments, analyzed the data, authored or reviewed drafts of the article, and approved the final draft.
- Caixia Zhang conceived and designed the experiments, analyzed the data, authored or reviewed drafts of the article, and approved the final draft.

### Human Ethics

The following information was supplied relating to ethical approvals (*i.e.*, approving body and any reference numbers):

The study was approved by the ethical committee of the School of Public Health of Sun Yat-Sen University (approval number: 2023-007).

### Data Availability

The raw data used for analysis is available in the Supplemental File.

### Supplemental Information

Supplemental information for this article can be found online at http://dx.doi.org/10.7717/peerj.17203#supplemental-information.

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
