# Peer review of "Association of healthy lifestyle score with control of hypertension among treated and untreated hypertensive patients: a large cross-sectional study"

_PeerJ, doi:10.7717/peerj.17203_

## Round 0.1 · original submission · Major Revisions

We appreciate the opportunity to review the manuscript. Our careful evaluation has led us to the decision that, while the research is promising, substantial revision is required before considering it for publication. Our primary concerns revolve around the methodology and results sections.

1- Introduction and literature review
The introduction and literature review must be expanded to incorporate recent research on the association between a healthy lifestyle score and hypertension (as well as other disease). This will give the reader an idea about what we know about the topic. A swift search has yielded at least 6-10 pertinent articles about HTN and lifestyle. This addition will strengthen the theoretical foundation and better position the manuscript within the existing body of knowledge.

2- Methodology and results
a. The methodology requires a more comprehensive explanation, particularly in relation to the comparison between individuals on treatment and those who are not. The results should be restructured to highlight this novelty, with a focus on statistical analyses specific to this question.

b. The lifestyle score is a critical aspect, and while we acknowledge the absence of a validated tool, the selection of variables and their measurement must be clarified. Justification for the inclusion of each item and the chosen measurement methods should be explicitly outlined. I agree with the reviewer that the use of both waist circumference (WC) and body mass index (BMI) is not advisable. Please address this concern and provide a clear rationale for the inclusion of two variables for one aspect of lifestyle. Please include the justification of the operation of each variable. A clear justification of the cut off points of the healthy lifestyle score is also needed.

Discussion:
The discussion section should go beyond a mere reiteration of results. It should offer insights for future research and practical implications of the findings. Highlight avenues for further investigation and consider the broader implications of the study's outcomes.

Reviewer 1 ·

Basic reporting

1. The language is good enough for review but should be improved for publication.
2. The study aimed to investigate the association between HLS and blood pressure control among treated and untreated hypertensive adults, but no data was provided in such a way in the Abstract. And the difference in the assoications between treated and untreated paptients was unknown.

Experimental design

1. As previously commented, the difference in the association of HLS with BP control between treated and untreated patients should be statstically tested, in order to conclude on the current study question.
2. Both WC and BMI are measuring obeisity. For scoring the HLS, if both of them are included and separately given a score, the HLS score should be 'over' scaled on obesity. Because obesity is strongly associated with blood pressure, the association of HLS with BP control might be biased to a positive association. I suggest to use either one.
3. Table 1 should show data by treated/untreated patients, not controlled/uncontrolled patients.
4. Table 2 should keep only data among all patients and data for treated and untreated patients.
5. The two boxes at the bottom of Figure 1 should be replaced with "treated" and "untreated" patients.
6. Figure 3a and Figure 3b should be deleted.

Validity of the findings

1. In the Abstract, the sentence 'The HLS was defined by six low-risk lifestyle factors: healthy dietary habits, sufficient physical activity, ideal body mass index, suitable waist circumference, non-smoking, and no alcohol consumption." is not clear enough. For example, what deitary habits were defined 'healthy'. And the scale of HLS was not given.
2. In the abstract, “Hypertensive patients who adhered to a healthy lifestyle without taking medication had better blood pressure management than those who used medication and followed a healthy lifestyle.” is lacking supporting evidence.

Reviewer 2 ·

Basic reporting

English is professional and clear, and the motivation and research goals are well-stated.

Experimental design

Target population, study location, and the time element are well-defined in the method section.

Line 95-96: How different are these two groups of participants (with or without completed profile of blood pressure and lifestyle information)? Could you please justify why you decided to handle the missing information by excluding them from the study?

Line 96: Define extreme values.

Lines 123-125: How many times was the blood pressure measured? What device was used for the survey? I recommend providing more details about the blood pressure measurement.

Have any validation studies been conducted regarding the health lifestyle score? What is the impact of self-reporting responses on the accuracy of the health lifestyle score? How effectively was the score combined to represent the participants' lifestyle?

Lines 147-149: If possible, please reconsider the method for selecting variables for multivariable adjustment. Adjusting variables that are imbalanced between groups doesn’t ensure that biases were properly adjusted.

Line 154: Please define 'the interactive effect.' I am unsure if it's a commonly used term in epidemiology.

Validity of the findings

No comment

---

## Round 0.2 · Minor Revisions

Thank you for addressing the comments and redoing the analysis. However, I would appreciate it if you could address the minor comments given by the reviewer, especially about the missing data and any potential bias. Have you done data on the complete set? have you compared the demographics of those with missing values? is it possible to include that as potential bias and limitation to the study.

Reviewer 1 ·

Basic reporting

no comment.

Experimental design

no more comment.

Validity of the findings

no comment.

Additional comments

No more comment.

Reviewer 2 ·

Basic reporting

The introduction section is well-articulated and supported by sufficient evidence. It flows nicely, and the study's objective is clearly presented.

Experimental design

Line 107: What are the differences in characteristics when comparing participants with or without missing data? Could potential biases be introduced by the complete case analysis, which is not currently recommended?

Validity of the findings

Line 241: I would suggest changing the wording from 'taking medication' to something like self-reported medication use for clarity. The survey might not accurately determine whether participants are actually taking medications.

The study's limitations are comprehensive. My concern is that the HLS score does not seem to have been previously validated in terms of how well it captures information about people's lifestyles, especially when BMI is the only verifiable piece of information in the score.

---

## Round 0.3 · Minor Revisions

Please include the table 1 mentioned in the response letter in the manuscript as a supplement as suggested by the reviewer. You can refer to the supplement in the following statement: Page 14, Lines 288-290. It reads as follows “Firstly, a large number of participants were excluded due to incomplete data, which could lead to selection bias and impact the generalizability of our findings”.

Reviewer 2 ·

Basic reporting

NA

Experimental design

NA

Validity of the findings

The authors have appropriately addressed my comments by commenting in the discussion section regarding the limitations of the research. If possible, please add the table presenting the differences in characteristics between participants with or without missing data in the manuscript as a supplement to provide additional information about which population this study may apply to.

---

## Round 0.4 · accepted · Accept

Thank you for your patience and cooperation in addressing all the reviewers' comments. The manuscript is ready for publication.